# Transfer-Learning-Based Approach for Yield Prediction of Winter Wheat from Planet Data and SAFY Model

Yu Zhao [1,2], Shaoyu Han [2], Yang Meng [1,2], Haikuan Feng [1,2], Zhenhai Li [2,3], Jingli Chen [4], Xiaoyu Song [2], Yan Zhu [1] and Guijun Yang [1,2,*]

1 National Engineering and Technology Center for Information Agriculture, Nanjing Agricultural University, Nanjing 210095, China
2 Key Laboratory of Quantitative Remote Sensing in Agriculture of Ministry of Agriculture and Rural Affairs, Information Technology Research Center, Beijing Academy of Agriculture and Forestry Sciences, Beijing 100097, China
3 College of Geomatics, Shandong University of Science and Technology, Qingdao 266590, China
4 College of Agriculture, Qingdao Hengxing University of Science and Technology, Qingdao 266100, China
* Correspondence: yanggj@nercita.org.cn

**Abstract:** Crop production is one of the major interactions between humans and the natural environment, in the process, carbon is translocated cyclically inside the ecosystem. Data assimilation algorithm has advantages in mechanism and robustness in yield estimation, however, the computational efficiency is still a major obstacle for widespread application. To address the issue, a novel hybrid method based on the combination of the Crop Biomass Algorithm of Wheat (CBA-Wheat) to the Simple Algorithm For Yield (SAFY) model and the transfer learning method was proposed in this paper, which enables winter wheat yield estimation with acceptable accuracy and calculation efficiency. The transfer learning techniques learn the knowledge from the SAFY model and then use the knowledge to predict wheat yield. The main results showed that: (1) The comparison using CBA-Wheat between measured AGB and predicted AGB all reveal a good correlation with $R^2$ of 0.83 and RMSE of 1.91 t ha$^{-1}$, respectively; (2) The performance of yield prediction was as follows: transfer learning method ($R^2$ of 0.64, RMSE of 1.05 t ha$^{-1}$) and data assimilation ($R^2$ of 0.64, RMSE of 1.01 t ha$^{-1}$). At the farm scale, the two yield estimation models are still similar in performance with RMSE of 1.33 t ha$^{-1}$ for data assimilation and 1.13 t ha$^{-1}$ for transfer learning; (3) The time consumption of transfer learning with complete simulation data set is significantly lower than that of the other two yield estimation tests. The number of pixels to be simulated was about 16,000, and the computational efficiency of the data assimilation algorithm and transfer learning without complete simulation datasets. The transfer learning model shows great potential in improving the efficiency of production estimation.

**Keywords:** yield; satellite remote sensing; crop growth model; SAFY; transfer learning; aboveground dry biomass

## 1. Introduction

Crop yield is of great importance to food security and directly relates to the economic benefits of farmers [1]. It is therefore an imperative task for robust large-scale yield estimation in agricultural research and application [2].

With the advantages of non-destructive, high throughput and spatially continuous observation, remote sensing (RS) is a popular method for monitoring bio-physical parameters [3–6]. Over the past decades, the application of RS information obtained by optical and radar sensors on the terrestrial, aerial and satellite platforms in estimating crop production have considerably increased, including empirical, semi-empirical and mechanism models. Compared with traditional image processing tools, the Google Earth Engine (GEE) platform facilitates RS data acquisition, RS image processing, and analysis [7]. The GEE platform

provides favorable conditions for crop yield prediction, crop growth monitoring and crop classification on a large regional scale [8–10]. For simplicity and computational efficiency, the majority of studies on crop yield prediction have been applied for crop yield estimation using vegetation indices (VIs) [11]. Some of the vegetation indices, i.e., the Normalized Difference Vegetation Index (NDVI) [12], the Enhanced Vegetation Index (EVI) [13] and the Optimized soil-adjusted vegetation indices (OSAVI) [14], are based on the Red-NIR isolines of the electromagnetic spectrum, prompted by the motivation to predict bio-physical parameters using bands available from different remote sensing platforms. With the future application of RS data in agriculture, multi-band VI, multi-calculation form and radar VIs are constructed [15]. The simplest method to estimate crop yield is to find the statistical relationships between only VIs and crop yield at specific phenological stages or multiple phenological stages, which could express acceptable crop yield estimation accuracy and performance under specific regions and years. The booting stage and flowering stage are often considered be the most suitable period to predict crop yield using optical remote sensing data [6,16]. Poor Spatial-temporal extrapolation without recalibration and revalidation confines the further application of these methods. The yield formation process is affected by many factors, such as soil, meteorology and genes [17–19]. Typically, this is now done by using different methods, e.g., hierarchical linear model [6] and machine learning methods [5,20], to fuse RS data with information related to crop growth. By integrating hyperspectral data and meteorological data, Li et al. [6] developed hierarchical linear modeling (HLM) to solve the interannual expandable problem of the wheat yield prediction model. To better explain the response of crop yield to changes in sub-seasonal environmental factors across the entire growing season, many machine learning methods and phenological information were considered for yield estimation models [5,21,22]. Despite the excellent performance of a semi-empirical, a well-trained model depends on limited mechanisms of crop growth that still have defects in expressing the process of crop yield formation, and their generality for multiple regions has not been thoroughly validated.

The process-oriented crop growth model (CGM) is a verified tool to simulate crop development and crop yield formulation at a farm scale with detailed input data, e.g., crop varieties, agronomic management practices, soil, and climate factors [22–24]. As a bridge between CGMs and RS data, data assimilation technology combined with the advantages of RS data and CGMs has been recognized as a reliable way to improve crop yield simulations over a large area. Crop growth variables are selected for assimilation to adjust the CGM simulation and thus obtain updated crop yield predictions. Among all the variables available for assimilation, leaf area index, aboveground dry biomass, canopy cover and soil moisture have been popularly elected in the assimilation system [25–28]. Aboveground dry biomass (AGB) is strongly related to crop yield, consequently, statistical regression, machine learning method and radiation transfer model have all been used to monitor AGB. Active and passive RS often has the ability to better predict AGB at low coverage, but all face the difficulty of saturation at high biomass [29]. Multi-source input variables are used in AGB prediction models, including optical VIs and SAR indicators, optical VIs and imagery textures, and optical VIs and phenological information [30–32]. Therefore, an in-depth study on agronomic parameters prediction using RS data is of great significance to further enhance crop yield prediction performance in the data assimilation system. In general, the estimation of agronomic parameters using the combination of multi-source data is more effective than that of only single-source data [33]. Different CGMs differ in their structure, parameterization and complexity, and may obtain different results even if the same observation data is applied [22–24]. Therefore, the data assimilation combined only one crop growth model, a yield estimation assimilation system using CGMs have also been constructed and achieved acceptable performance [34,35]. Although crop yield estimation using data assimilation has made great progress, problems remained. Primarily, for the periods of state variables extraction, they were not well represented, previously used state variables cannot represent the environmental factors resulting in the lack of universality in crop yield estimation under different environments and conditions. Second,

for the assimilation algorithms, the computational intensity was always a problem that could not be ignored when estimating production in a large area. Summarizing the existing issues above, the existing problem should be using sufficient data which can be mined from simulation results of CGMs and RS data to complete the estimation of crop pre-harvest yield with less calculation.

Recently, machine learning methods have been used to learn abstract features from high-spatial-resolution RS images for yield estimation [20]. Despite the excellent performance of machine learning methods, a trained network depends on the large number of labeled samples that are difficult and expensive to collect in agricultural applications. Therefore, an economical approach such as transfer learning techniques would be desirable for crop yield prediction for different environmental conditions. To date, there has been limited work done to construct transfer learning techniques to improve local crop yield predictions. Therefore, the major objectives were as follows in this study: (1) to explore dynamic patterns of aboveground dry biomass (AGB) observations and simulated crop yield to present a transfer learning yield inversion model; (2) to examine the performance and transferability analysis of transfer learning yield inversion model cross-year yield estimation; (3) test the computational efficiency and accuracy of the data assimilation system and the proposed method.

## 2. Materials and Methods

### 2.1. Study Location and Field Data Collection

Field experiments (40.17°N, 116.43°E) were conducted in 2017~2018 and 2018~2019 at National Precision Agriculture Research Center near Beijing, China (see Figure 1). The historical experiments involve different cultivars, nitrogen management and growing seasons [32]. Two experiments (Calibration sets), 2017–2018 (Exp. 1) and 2018–2019 (Exp. 2) were designed as completely randomized designs with cultivars (Jingdong 18 and Lunxuan 167) and four nitrogen levels (N1: 0 kg N ha$^{-1}$; N2: 90 kg N ha$^{-1}$; N3: 180 kg N ha$^{-1}$, and N4: 270 kg N ha$^{-1}$). There were 32 plots per growing season, with 128 biomass samples (32 plots × 4 periods) and 32 yields collected annually, see Li et al. [32] for details. At each sampling site, Zadok's growth stage [36] was investigated. Additionally, a yield survey was performed inside the farm in 2018 and 2019, 32 and 71 yield samples were randomly collected respectively for verification of the yield estimation model (Figure 1b).

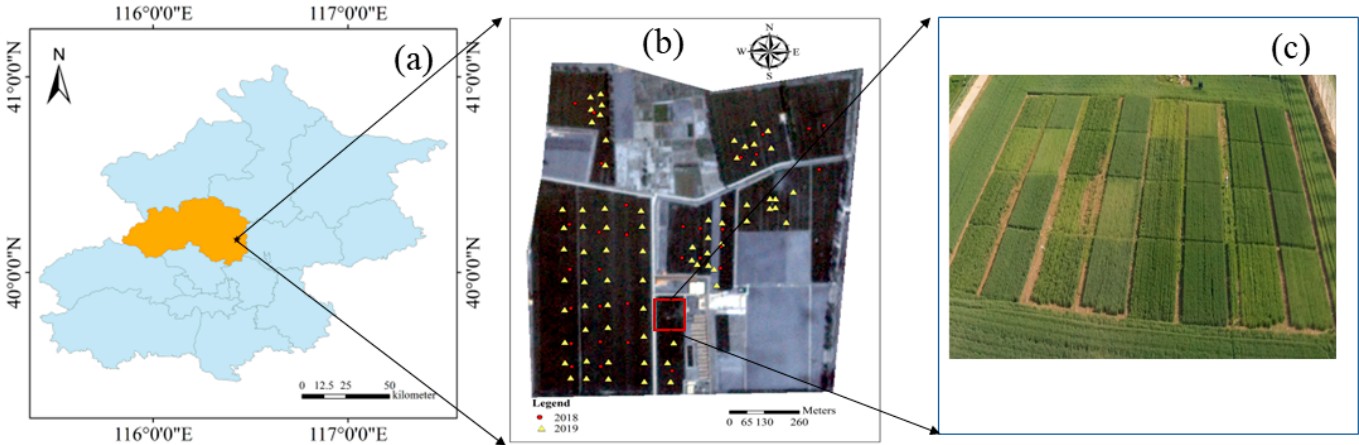

**Figure 1.** Experimental farm location for field observations: (**a**) test site; (**b**) validation set area; (**c**) calibration set area. Orange area in Figure 1a represent the Changping district.

### 2.2. Data Collection

2.2.1. Satellite Imagery Data Acquisition and Preprocessing

Planet satellite (www.planet.com, accessed on 1 September 2022) is an emerging remote sensing satellite in 2018~2019, with high frequency, good image quality and high

data coverage efficiency. There are more than 170 satellites in the Planet's small satellite constellation. Hundreds of satellites independently photograph the world every day, which can achieve daily global coverage. The Planet satellite imagery parameters are shown in Table 1. The images used in this paper are 3B-level data products after sensor calibration, radiometric calibration, orthorectification and atmospheric correction. The detailed process of satellite data processing is shown in Bai et al. [37].

**Table 1.** Planet satellite remote sensing data parameters.

| Bands | Wavelength (nm) | Spatial Resolution (m) |
|-------|-----------------|------------------------|
| Blue | 455–515 | 3 |
| Green | 500–590 | 3 |
| Red | 560–670 | 3 |
| NIR | 780–860 | 3 |

A commonly used vegetation index from literature, the enhanced vegetation index 2 (EVI2) [38], was used to predict AGB in this study. The formula is as follows:

$$EVI2 = 2.5 \times (NIR - R)/(NIR + 2.4 \times R + 1) \tag{1}$$

where 2.5, 2.4 and 1 are the adjustment factors to reduce the impact of soil, atmosphere and saturation on the prediction of agricultural parameters. NIR represents the reflectance of the near-infrared band and R represents the reflectance of the red band.

### 2.2.2. Meteorological Data

The meteorological data from the ERA5 dataset (ECMWF, http://www.ecmwf.int, accessed on 1 September 2022) was used in this study, including daily minimum temperatures (Tmin), maximum temperatures (Tmax) and solar radiation, were acquired with a resolution of 0.125° for the experimental site. The daily meteorological data in this study are shown in Figure 2.

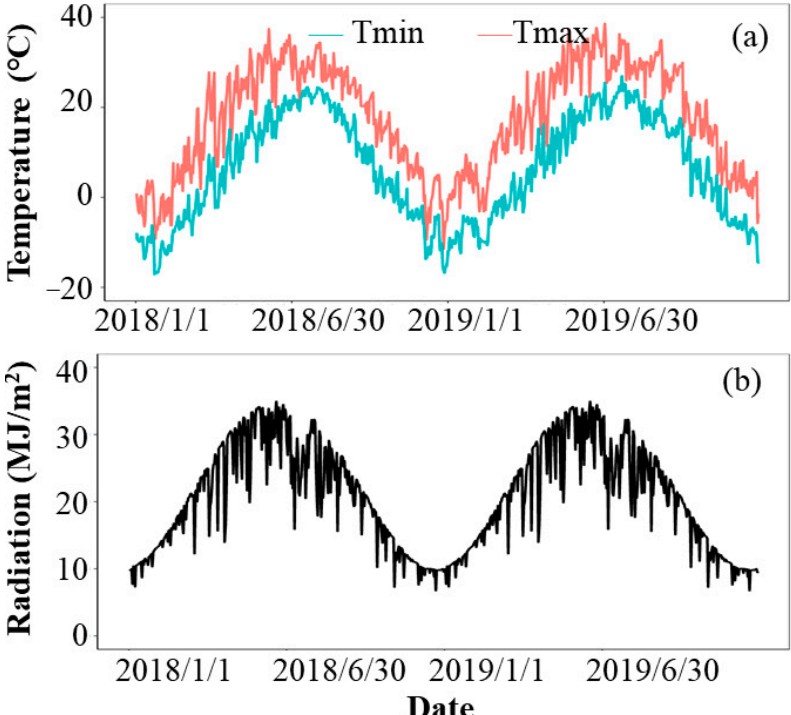

**Figure 2.** Daily temperature (**a**) and radiation (**b**) during 2017~2019 in field experiments.

### 2.2.3. Field Measured AGB

In each growth period, the wheat tillers number per unit area was investigated in each experimental plot. Destructive sampling of 20 randomly selected tillers were taken to a laboratory and 20 tillers samples in each sampling point were then oven-dried to a constant weight. The wheat AGB (t ha$^{-1}$) was calculated as follows:

$$AGB = \frac{W \times T}{S \times A} \times 10^{-2} \tag{2}$$

where S represents the number of tillers in the destructive samples, W represents the weight (g) of the destructive samples, A represents the sampled area (m$^2$), and T represents the total number of counted tillers in the sampled area (1 m$^2$ in this study). The $10^{-2}$ represent the coefficient transformed from g m$^{-1}$ to t ha$^{-1}$.

### 2.2.4. Winter Wheat Yield Observations

A 1 m$^2$ area of the winter wheat was destructive sampled to measure the harvested yield. Each sample point was in the center area of the experimental plot to avoid edge effects. The harvested grain had a 14% moisture content was threshed, air dried, and weighed on an electronic balance.

### 2.3. Data Analysis

In this study, one transfer learning method for yield prediction based on the Simple Algorithm For Yield (SAFY) model (see Section 2.3.1) [39], and the deep neural network (DNN) [40] method was constructed. Data assimilation using Shuffled complex evolution with PCA (SP-UCI) [41,42] by integrating the remote sensed AGB and SAFY model was tested in comparison to the capacity of the constructed model (transfer learning method). A detailed flowchart for the yield estimation method is presented in Figure 3.

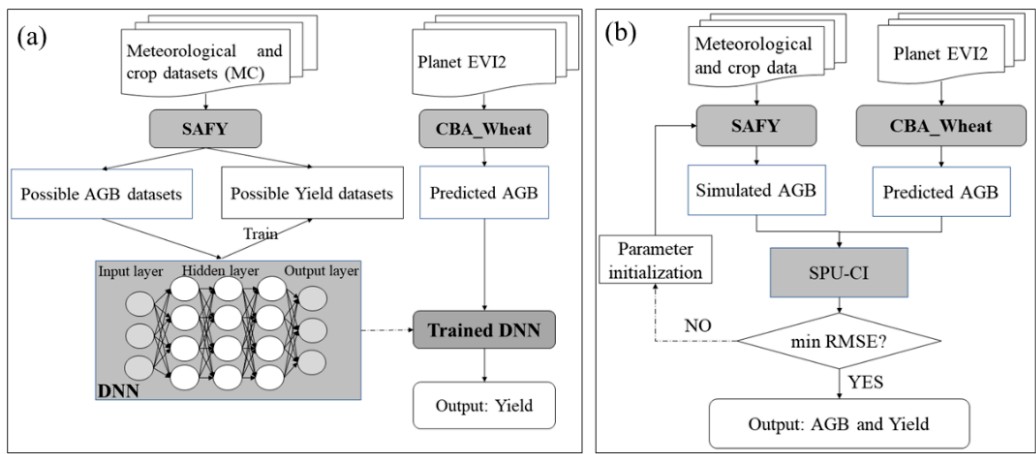

**Figure 3.** The flowchart of transfer learning method (**a**) and data assimilation (**b**) for winter wheat. SAFY, CBA_Wheat, DNN, SPU-CI, AGB and EVI2 represent the simple algorithm for yield model, crop biomass algorithm for wheat, the deep neural network, the Shuffled complex evolution with PCA, aboveground dry biomass and the enhanced vegetation index 2, respectively.

### 2.3.1. Principle of SAFY Model

SAFY, a simple crop growth model, is used for simulating AGB on a daily step and is easy to integrate RS data for estimating AGB and crop yield [3,43,44]. During the AGB increased period, daily AGB production is based on light-use efficiency (LUE) theory, depending on effective LUE (ELUE), photosynthetically active radiation absorbed by crop

canopy (APAR), daily average temperature (Ta) and temperature-stress-function ($F_T$) [40]. Which uses the following equation:

$$\Delta AGB = ELUE \times F_T(Ta) \times APAR \tag{3}$$

$$APAR = \varepsilon_1 \times \varepsilon_c \times Rg \tag{4}$$

where, $\varepsilon_1$, $\varepsilon_c$ and Rg represent light-interception efficiency, climatic efficiency and daily incoming global variation. Where, $\varepsilon_1$ depends on light-interception coefficient (k) and green leaf area index (GLAI) based on Beer's law:

$$\varepsilon_1 = 1 - e^{-k \times GLAI} \tag{5}$$

The daily maximum and minimum temperature directly affect the daily AGB accumulation rate, which is accounted for by Ta in the SAFY model determined by minimal temperature (Tmin), maximal temperature (Tmax), and optical temperature (Topt) for growth. Which uses the following equation:

$$F_T(Ta) = \begin{cases} 1 - \left[\frac{Topt-Ta}{Topt-Tmin}\right]^2, & \text{if } Tmin < Ta < Topt \\ 1 - \left[\frac{Ta-Topt}{Tmax-Topt}\right]^2, & \text{if } Topt < Ta < Tmax \\ 0, & \text{if } Ta < Tmin \text{ or } Ta > Tmax \end{cases} \tag{6}$$

The dynamics of GLAI can be split into two phenological phases, leaf extent expansion and leaf disappearance during crop senescence, which is determined by the sum of temperature (STT) after crop emergence. In the leaf expansion stage, both AGB and GLAI are in the growth stage, a portion of the daily accumulated AGB by allocation to leaf function (Pl) was partitioned to dry leaf biomass, and then the increase of daily dry leaf biomass is converted in increase of daily leaf area ($\Delta GAI^+$) based on specific leaf area (SLA). If Pl > 0, $\Delta GAI^+$ leads to:

$$\Delta GAI^+ = \Delta AGB * Pl * SLA \tag{7}$$

$$Pl = 1 - Pl_a * e^{Pl_b * SMT} \tag{8}$$

where $Pl_a$ and $Pl_b$ represent two partition-to-leaf parameters and SMT represents temperature accumulation. When the SMT reached the given threshold STT, the leaves start to senesce at a constant rate (Rs). If SMT > STT, it leads to:

$$\Delta GAI^- = GLAI \times \left(\sum Ta - STT\right)/Rs \tag{9}$$

During the grain filling phase, the crop yield is proportional to the maximal AGB, with a constant harvest index (HI) partitioned to grains. It leads to:

$$Yield = HI \times AGB_{max} \tag{10}$$

Parameters of the SAFY model such as GLAI and ELUE in the model indirectly characterize the simulated yield differences caused by other agro-environmental stress. The description and range of parameters of the SAFY model are as follows (see Table 2):

**Table 2.** Parameter descriptions in the SAFY model.

| | Parameter | Abbreviation | Unit | Value | References |
|---|---|---|---|---|---|
| Fixed | Climatic efficiency | $\varepsilon_c$ | - | 0.48 | Duchemin et al. [39] |
| | Light-interception coefficient | $\varepsilon_1$ | - | 0.5 | Duchemin et al. [39] |
| | The optimal temperature | Topt | °C | 21 | Wang et al. [45] |
| | Minimum temperatur | Tmin | °C | 0 | Wang et al. [45] |
| | Maximum temperatur | Tmax | °C | 37 | Wang et al. [45] |
| | Specific leaf area | SLA | $m^2/g$ | 0.022 | Claverie et al. [46] |
| | Initial aboveground biomass | $AGB_0$ | $g/m^2$ | 4.5 | Calibrated |
| | Leaf Partitioning Coefficient a | Pla | - | 0.16 | Calibrated |
| | Leaf Partitioning Coefficient b | Plb | - | 1.4 | Calibrated |
| | Senesce rate | Rs | °C/d | 10.8 | Calibrated |
| Calibrated | Day of emergence | $D_0$ | d | 0–15 | This study |
| | Sum of temperature | STT | °C | 1200–1600 | This study |
| | Effective Light Use Efficiency | ELUE | g/MJ | 1.3–2.5 | This study |

### 2.3.2. Predicted AGB from Planet Imageries Using the CBA-Wheat Model

Most existing AGB monitoring models perform well only at a specific growth stage at a single location, with poor transfer during different growing stages during the season [28,43]. AGB is an important feature of the SAFY model, and excellent AGB estimation is of great significance for transfer learning to learn the characteristics of the SAFY model for yield estimation. The newly developed crop biomass algorithm for wheat (CBA-Wheat) [28] was chosen to predict winter AGB over the growing season, with a two-level piecewise model, such as:

$$AGB = \alpha \times EVI2 + \beta \tag{11}$$

$$\alpha = 1.27e^{p1 \times ZS} \tag{12}$$

$$\beta = p2 \times ZS - p3 \tag{13}$$

where $\alpha$ and $\beta$ represent the CBA-Wheat coefficients, ZS represent Zadok's growth stage [37]. p1, p2 and p3 are the CBA-Wheat models constructed based on Li et al. [32]. Although this paper uses the same sets of ground data as Li et al. [32], considering the differences between the Planet satellite and ground hyperspectral data, the coefficients are fine-tuned.

### 2.3.3. Transfer Learning Method

The general framework of the transfer learning method (see Figure 3a) in this study is to summarize a crop growth model (SAFY) to use the information in the simulated labeled AGB-yield datasets to estimate the wheat yield and improve the efficiency of yield estimation. The transferability of features extracted by deep neural networks has been demonstrated and applied to natural language processing, computer vision and other fields [47–49]. This study then chose root mean squared error (RMSE) and loss value to evaluate the model accuracy. The implementation steps of the transfer learning method are as follows:

(I). SAFY parameter sets construction: 200,000 sets of parameter combinations of SAFY were generated based on Monte Carlo (MC) algorithm.

(II). Construction of possible AGB datasets and yield datasets based on the SAFY model: The parameter set constructed in step I was input into the SAFY model to obtain the possible AGB datasets and yield datasets. The time efficiency test of transfer learning is divided into two types with or without simulated data sets: no simulated datasets and with simulated datasets.

(III). Train DNN model: A four-layer fully-connected network is constructed to train simulated AGB-yield from the SAFY model. The model was pre-trained using the AGB datasets and yield datasets simulated in step II, and fine-tuned by the measured data using transfer learning.

(IV). Forecast yield based on transfer learning method: The AGB predicted from the CBA-Wheat model is utilized as the input layer of the trained DNN to predict winter wheat yield.

2.3.4. Data Assimilation

In this section, the Shuffled complex evolution with PCA (SP-UCI) algorithm is improved based on the SCE-UA algorithm and combines compound evolutionary algorithm, simplex algorithm and polynomial resampling. SP-UCI not only maintains the searchability of particle swarm in the whole parameter space but also efficiently realizes high-dimensional parameter optimization [41,50], and it is useful and effective global optimization. By using the SP-UCI algorithm, three sensitive parameters such as $D_0$, ELUE and STT were optimized, and then the dynamic growth simulation was performed for the entire growth period of winter wheat using the SAFY model. The implementation steps of the data assimilation method (Figure 3b) are as follows:

(I). AGB predicted model construction: the AGB prediction results based on the CBA-Wheat model are chosen as the state variable to estimate the yield in the assimilation system.

(II). Run SAFY: SAFY model is run based on initialized model parameters and meteorological data.

(III). Cost function calculation: The cost function is built on the basis of the relationship between the measured AGB and the model simulated AGB.

$$\text{RMSE} = \sqrt{\frac{1}{n}\sum_{i=1}^{n}(Y_i - Y_i\prime)^2} \tag{14}$$

where, n, $Y_i\prime$, $Y_i$, and p are the number of samples, predicted AGB value, measured AGB value, and the number of independent variables, respectively.

(I). Determine iteration termination conditions: When the objective function cannot be improved by 0.01% or the cost function is calculated more than 10,000 times to terminate the cycle.

(II). Test the error between the model measured yield and the simulated yield.

## 3. Results

### 3.1. Validation of AGB Retrieved from CBA-Wheat Model and SAFY Model

Figure 4a shows that the trend of simulated AGB could reflect almost field-measured AGB in all nitrogen treatments. Four phases including the jointing phase, booting phase, flowering phase and filling phase were picked out. The distribution of AGB in the measured dataset, possible datasets, CBA-Wheat and parameters data assimilation dataset were illustrated in Figure 4b from jointing phase to filling phase, which demonstrates the broad coverage of the possible datasets. Moreover, the shapes of the four kinds of AGB datasets were similar. In this paper, the range and mean value of measured AGB were 0.46~19.50 t ha$^{-1}$ and 6.96 t ha$^{-1}$, respectively. The AGB ranges from the possible datasets of the jointing phase, booting phase, anthesis phase and filling phase were 0.10~7.72 t ha$^{-1}$, 2.13~12.79 t ha$^{-1}$, 3.20~19.11 t ha$^{-1}$ and 5.24~26.36 t ha$^{-1}$, respectively. The range of these values included the AGB values of other data sets in the corresponding period, indicating that the datasets input the transfer learning model include all possible situations of field production.

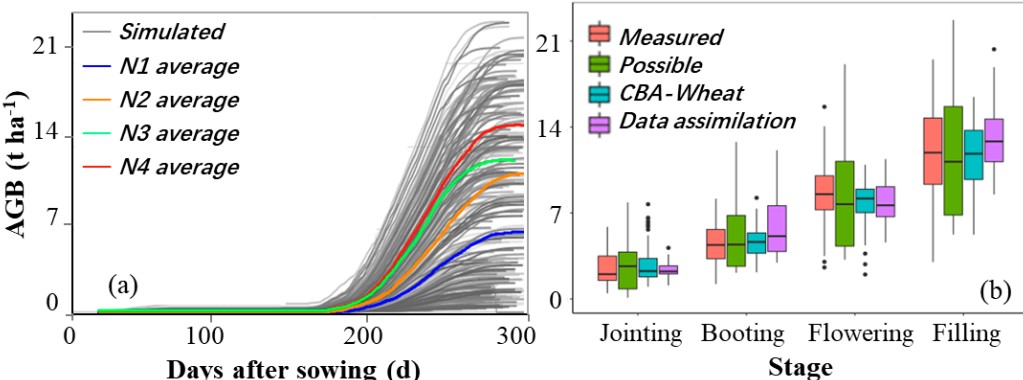

**Figure 4.** Winter AGB distribution in different stages. (**a**) the measured and the simulated AGB in four different stages. (**b**) the AGB distribution in the measured dataset, the possible dataset from the SAFY model (Section 2.3.3), CBA-wheat dataset and data assimilation of the four growth phases.

### 3.2. AGB Distribution for Different Datasets in Different Stages

The predicted AGB values were compared to 384 destructive measured AGB during four growth phases. The comparison between the measured AGB and the predicted AGB using CBA-Wheat (Figure 5) reveals a good correlation with $R^2$ of 0.83 and RMSE of 1.91 t ha$^{-1}$. CBA-Wheat and data assimilation have similar AGB estimation accuracy at different growth stages, which provides a solid foundation for the transfer learning method to replace the crop growth model for yield estimation.

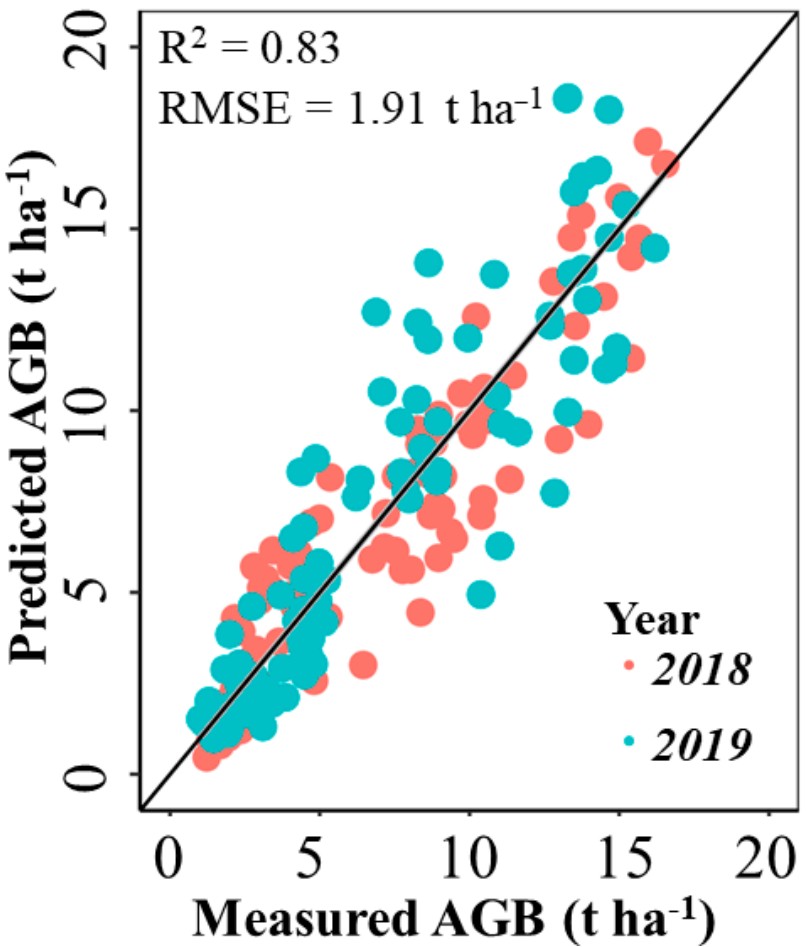

**Figure 5.** Comparison of measured and predicted AGB with CBA-wheat model.

### 3.3. Winter Wheat Yield Prediction

The simulated dataset with 60,000 samples covering the yield with $0.06\sim2.49$ t ha$^{-1}$ generated from the SAFY model was used to validate the effectiveness in yield inversions. The RMSE and loss value of the simulated dataset versus epochs are shown in Figure 6. When the training Epoch was more than 52, RMSE and loss value was less than 1.44 t ha$^{-1}$ and 0.03, respectively. Meanwhile, the values converged to stable as the training epochs increased. Figure 7 demonstrates the high consistency between the measured and the predicted yield derived from the data assimilation and transfer learning method. The wheat yield prediction performance was as follows: transfer learning method (Figure 7a: $R^2$ of 0.64, RMSE of 1.05 t ha$^{-1}$) and data assimilation (Figure 7b: $R^2$ of 0.64, RMSE of 1.01 t ha$^{-1}$).

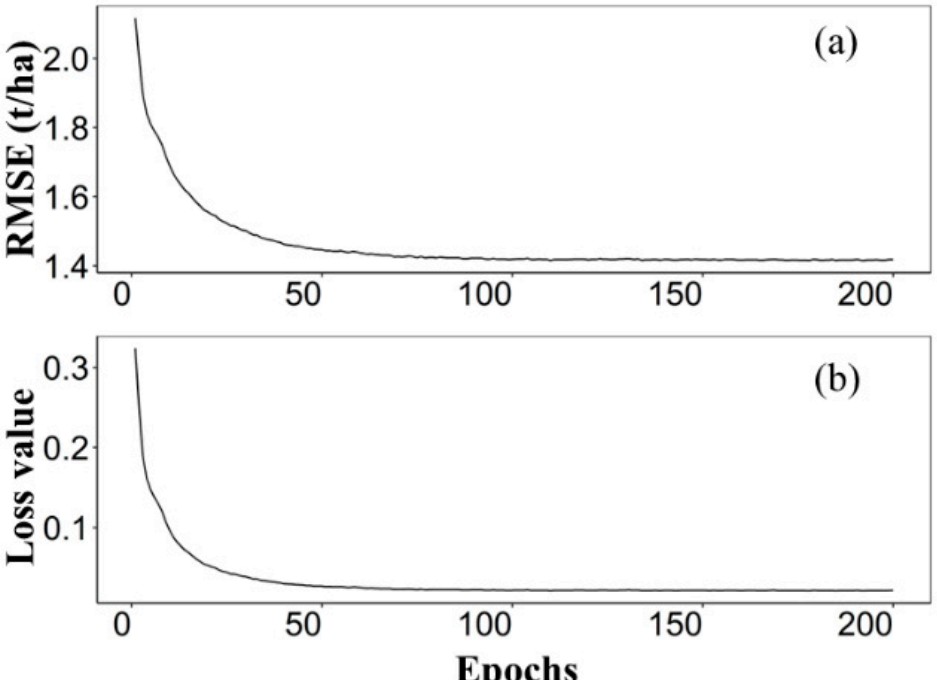

**Figure 6.** Performances of yield estimation model built using possible dataset and DNN: (**a**) RMSE, (**b**) loss value. RMSE, DNN and epochs represent root mean square error, the deep neural network and DNN iteration number, respectively.

To compare the computational efficiency of the transfer learning method and data assimilation algorithm, three supplementary experiments are designed (Figure 8).The time consumption of transfer learning with complete simulation datasets was significantly lower than that of the other two yield estimation tests. When the number of pixels was about 16,000, the calculation efficiency of the data assimilation system was the same. When the number of pixels is more than 16,000, the transfer learning without complete simulation datasets had better computational efficiency compared with data assimilation system.

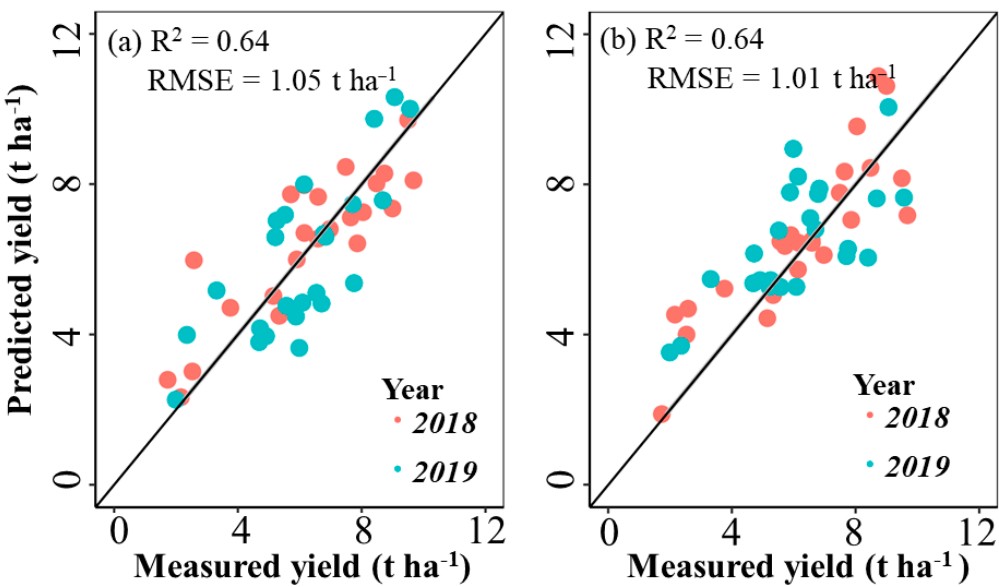

**Figure 7.** Scatter plots of measured yield versus predicted yield (tons per hectare): transfer learning method (**a**) and data assimilation (**b**).

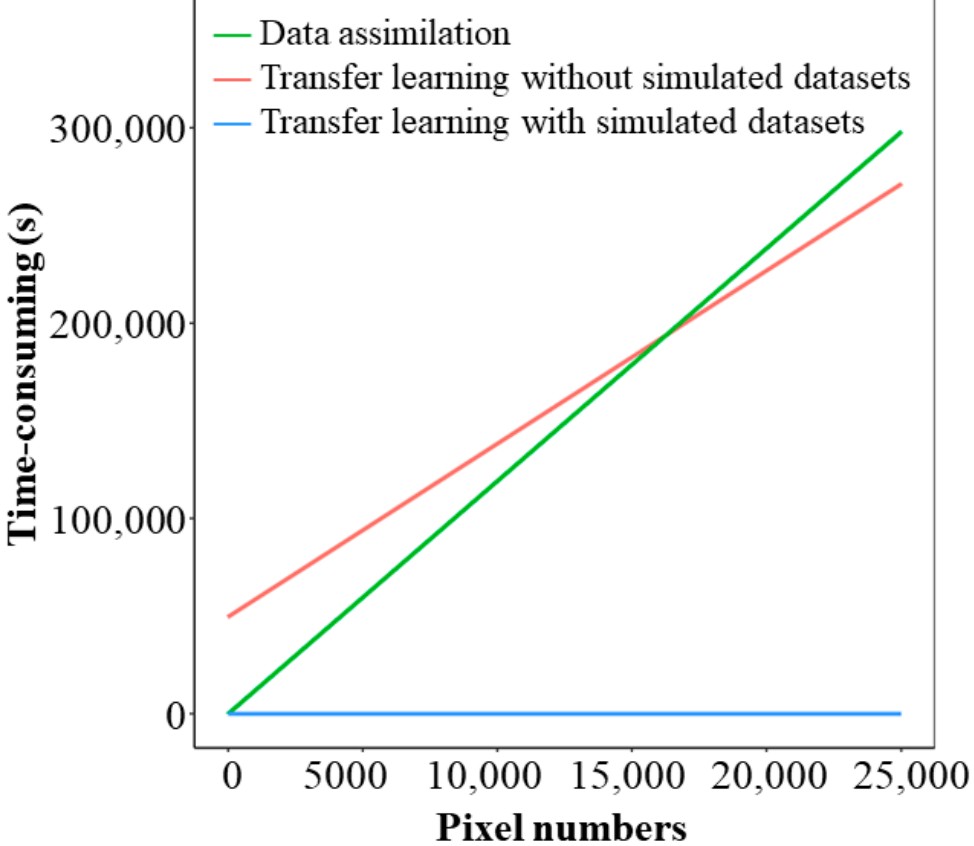

**Figure 8.** Comparison of computational efficiency between data assimilation and transfer learning algorithms. Computer configuration: Lenovo Legion R7000 2021 CPU R7-5800H @ 3.20 Hz and 16G random-access memory.

### 3.4. Farm-Land Verification of Transfer Learning Method

The SAFY and transfer learning method were verified on the farmland. The results of estimating yield with the data assimilation and transfer learning method are shown in Figure 9. At the farm scale, the two yield estimation models were still similar in performance for data assimilation (RMSE of 1.33 t ha$^{-1}$ and $R^2$ of 0.46) and for transfer learning (RMSE of 1.13 t ha$^{-1}$ and $R^2$ of 0.47). The farm-scale yield results (Figure 10) were consistent with the actual farm production. The above-mentioned results illustrated that, the transfer learning technique had good performance in calibration datasets and validation datasets. The transfer learning method, i.e., the DNN model, could exploit full merit of the prior knowledge extracted from the crop growth model to reach favorable yield predicting performance. Moreover, the generalization of the transfer learning method could also provide a potential insight for timely monitoring of wheat growth in the field.

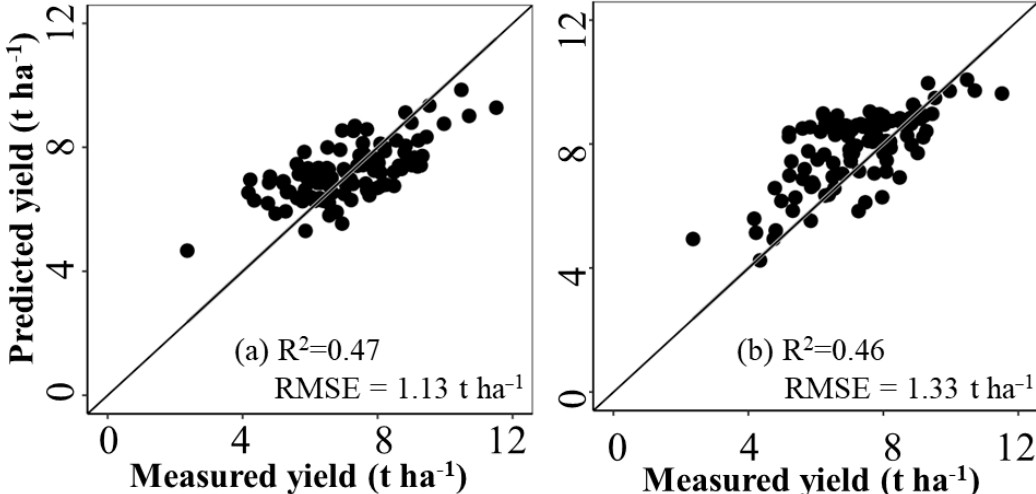

**Figure 9.** Predicted yield (tons per hectare) derived from transfer learning models (**a**) and data assimilation (**b**).

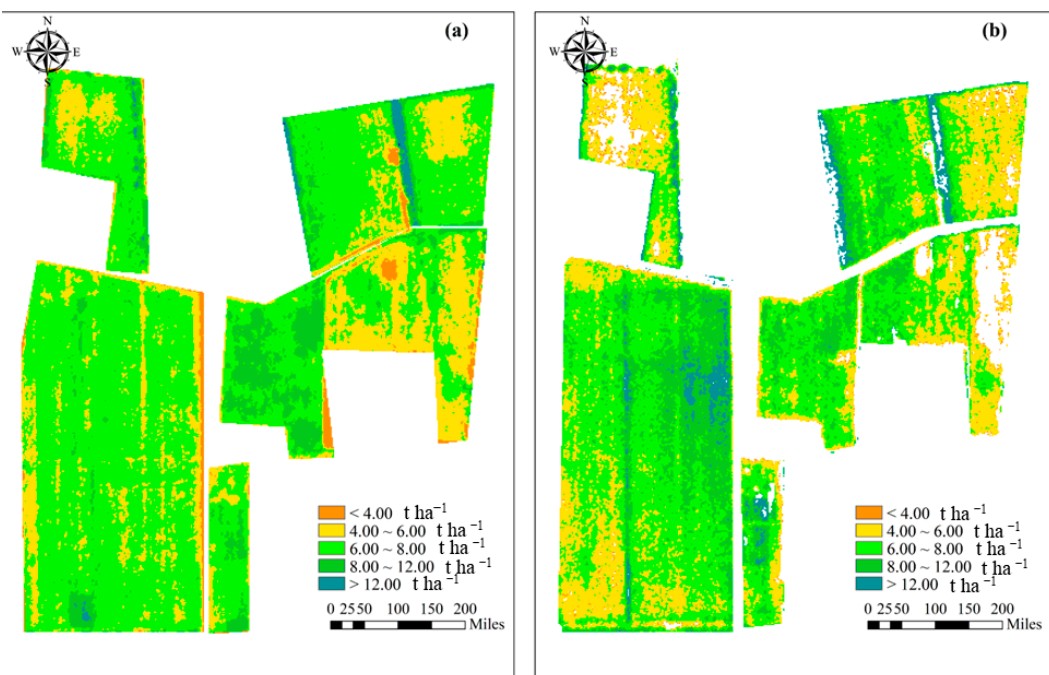

**Figure 10.** Winter wheat yield maps over experimental seasons: (**a**) 2018 and (**b**) 2019 at Xiaotangshan derived using Planet satellite imagery with the transfer learning models.

## 4. Discussion

Previous studies have attempted to predict yield using optical remote sensing technology [5,20,21]. The present study proposed a novel methodology for crop yield estimation based on satellite imagery. The advantages and limitations of the transfer learning method would be discussed from three aspects: the advantage of applying CBA-Wheat to predict AGB, the comparison between the transfer learning method and data assimilation and potential extension and limitation.

### 4.1. Advantage of Applying CBA-Wheat to Predict AGB

The AGB level at different growth stages has an obvious impact on the increase or decrease of harvest yield because crop yield is part of AGB. Different relationships of AGB-EVI2 at various growing phases have previously been reported in Roth., [51] and Li et al., [32], hence this was a widespread problem. On the one hand, the difference in organic matter accumulation rate and vegetation spectral response with the growth period resulted in different AGB changes corresponding to the change of unit vegetation index in different growth periods [32,52]. On the other hand, the limited canopy information that can be detected by the optical vegetation index limits the application of AGB estimation in multiple growth periods [53,54]. To eliminate the phenomenon of appeal, plant height, phenological information, vertical distribution and multi-angle optical information have been used in the modified AGB unified model of the multi-growth period. CBA-Wheat is a model aiming at the fact that VI cannot be directly used for AGB estimation in different growth periods. Compared with the method of integrating other information with the vegetation index, CBA-Wheat is simple, efficient and accurate, and can meet the accuracy of transfer learning. In this paper, EVI2 is used to estimate biomass and yield. EVI2 has the advantages of sensitivity and anti-saturation at a high biomass level, and the band of EVI2 is included in the existing satellite data [38]. However, EVI2 is built based on optical satellites and is vulnerable to clouds, rain and other meteorological factors. When using the model in a larger area, it is necessary to consider the fusion of multi-source data (such as combining active remote sensing data, etc.).

### 4.2. Comparison between Transfer Learning Method and Data Assimilation

Previous studies demonstrated the capacity of using data assimilation to improve crop productivity estimation for a large area [3,40,45]. In this research, instead of many complex crop growth models such as the Decision Support System of Agrotechnology Transfer modeling system (DSSAT), the Agricultural Production Systems sIMulator (APSIM), we chose only one mechanically simpler model (SAFY model) [40,55,56]. The reason for using the SAFY model is its advantages of requiring fewer detailed inputs, and capable of producing similar outputs compared with the complex models [57,58]. With optimization of tuning cultivar parameters, the SAFY model was able to predict the yield interannually, with a high $R^2$ of 0.64 to measured yield (Figure 7). The main reason for limiting its application can be attributed to low computational efficiency. Although the statistical method is highly efficient, it is not considered in this paper due to poor mechanism. The DNN network was trained on the AGB and yield generated from SAFY, then transferred to the field winter wheat yield prediction tasks. The proposed method has relatively simple architecture than data assimilation system. The yield prediction accuracies reached satisfactory results in the validation under various circumstances, including different locations and across years. The accuracy of the transfer model proposed in this study is similar to that of the data assimilation algorithm in yield prediction. Crop growth models are built according to the crop growth and development principles, this fact makes it possible to use SAFY simulated data as a base knowledge for winter wheat yield prediction [48]. However, the crop growth model is generality used to simulate crop growth and development under normal climate and management conditions, without considering extreme climatic conditions and pests [40,55,56]. Therefore, learning more crop growth models with different structures will help to predict yield and other parameters under different growth conditions. On the

premise that the simulation data set has been trained well, the computational efficiency of transfer learning with simulated datasets was much higher than that of a data assimilation algorithm (Figure 8). It can be explained that the transfer learning method can express more efficient results with fewer parameters [40]. At the image pixel number is about 16,000, the time-consuming of the data assimilation algorithm and transfer learning without complete simulation datasets is the same. Building usable simulation data sets is the most time-consuming step in transfer learning without complete simulation datasets. The turning point of computing efficiency will be different in different computer equipment and running software. Figure 10 showed higher yield was observed in the area close to the edge of the fields. On the one hand, the pixel at the edge of the field is a mixed pixel of grass and wheat, which may overestimate the yield; on the other hand, the wheat growing at the edge of the field had abundant in light, temperature and other environmental conditions, so it has a high organic matter accumulation. The remote sensing images used in this paper are mainly optical data, whose data quality is affected by clouds and rain, and then affects the yield prediction of the farm [23]. In the future, it is necessary to use multi-source data for large-scale regional yield estimation.

*4.3. Potential Extension and Limitation*

The results provided reliable evidence that crop yield could be predicted using the newly proposed transfer learning method with good accuracy and expansibility similar to data assimilation. Some potential extensions can be considered although more development is still needed. Massive simulation data based on the crop growth model is an important foundation of the hybrid method and provide representative information for later yield estimation. This study has only focused on winter wheat yield, other output parameters prediction of other crop growth models for different crops such as LAI, plant nitrogen concentration and crop quality, can be applied to enhance the estimation performance. The proposed hybrid method could enhance the use of simulation data generated by crop growth models, which would reduce the difficulty of manual sampling and data acquisition in the application of yield production in large regions. If the constructed sample set is poorly representative, the prediction performance of the migration model is still poor [59]. It worth highlighting that high-quality dataset is necessary, in order to apply transfer learning technology in practical agriculture and reach high yield estimation accuracy. Therefore, it is strongly required to formulate more scientific and standardized field sampling standards and establish a representative database, which will have a far-reaching impact on the development of yield prediction. Yield estimation from a farm was used to verify the effectiveness of this method. Although the transfer learning method was verified in different N management conditions, the verification of other treatments or other crop species was incomplete. Future research remains indispensable to evaluate the feasibility of such a method, for the extension when upscaling to different ecological areas. Whether a set of simulation datasets can be shared for transfer learning under different production conditions also needs to be further studied.

## 5. Conclusions

In this paper, to effectively utilize the satellite data to predict wheat yield, a transfer learning strategy was proposed, which included CBA-Wheat algorithm and the transfer learning method with SAFY simulation dataset and satellite-based observations. The method was fully analyzed, validated against field observations and compared with the data assimilation method, with the main conclusions as follows: (1) the comparison between measured AGB and predicted AGB using CBA-Wheat all reveal a good correlation with $R^2$ and RMSE of 0.83 and 1.91 t ha$^{-1}$, respectively. (2) the performance of yield prediction was as follows: transfer learning method ($R^2$ of 0.64, RMSE of 1.05 t ha$^{-1}$) and data assimilation ($R^2$ of 0.64, RMSE of 1.01 t ha$^{-1}$). At the farm scale, the two yield estimation models are still similar in performance with RMSE of 1.33 t ha$^{-1}$ for data assimilation and 1.13 t ha$^{-1}$ for transfer and learning. (3) The time consumption of transfer learning with complete

simulation data set is significantly lower than that of the other two yield estimation tests. While the number of pixels to be simulated is about 16,000, the computational efficiency of the data assimilation algorithm and transfer learning without complete simulation datasets.

**Author Contributions:** Conceptualization, Y.Z. (Yu Zhao), S.H. and Z.L.; methodology, Y.Z. (Yu Zhao), S.H. and Z.L.; software, Y.Z. (Yu Zhao); validation, Y.Z., S.H. and Y.M.; formal analysis, X.S. and G.Y.; investigation, G.Y. and Y.Z. (Yan Zhu); data curation, J.C., S.H. and Y.Z. (Yu Zhao); writing—original draft preparation, Y.Z. (Yu Zhao); writing—review and editing, Y.Z. (Yan Zhu); visualization, H.F.; funding acquisition, X.S., Z.L. and G.Y. All authors have read and agreed to the published version of the manuscript.

**Funding:** This research was funded by the earmarked fund for China Agriculture Research System (CARS-03), the National Natural Science Foundation of China (Grant No. 42271396) and the National Key Research and Development Program (2019YFE0125300).

**Data Availability Statement:** Not applicable.

**Acknowledgments:** The authors would like thank Weiguo Li, Hanzhong Liu and Hong Chang for acquiring data in the field experiments of this study.

**Conflicts of Interest:** The authors declare no conflict of interest.

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
