# Peer review of "Transfer-Learning-Based Approach for Yield Prediction of Winter Wheat from Planet Data and SAFY Model"

_remotesensing, doi:10.3390/rs14215474_

Round 1

Reviewer 1 Report

Dear Authors

Thank you for your manuscript. Please find it attached with comments. I had two minor concerns; thus you need to read through the manuscript and reconsider how you use the present and future tenses (was/is) etc. Be consitent throughout.  Secondly, there are numerous sentences without references. 

thank you 

Reviewer 2 Report

The manuscript aims to estimate wheat crop yield using remotely sensed biomass from EVI2 and the SAFY crop growth model. Two approaches, machine learning and data assimilation were used in the study. In general, the manuscript is clear to follow. The study is generally interesting.  The authors may consider several comments in the revision,

Figure 10, there may be some errors in the spatial pattern of crop yield. Highest crop yield was observed in the area close to the edge of the fields. Is that correct? In addition, the patterns of the estimated crop yield from the two years were quite different.  Explanations and discussions on this difference for the driving factors and uncertainties are required.

Discussions on the contribution of uncertainties/errors in the remotely sensed biomass from EVI2 on the crop yield estimation are required.  

Line 294, unclear about which approach used for the predicted AGB values.  Similar missing information through the whole manuscript are needed to be revised.

Figure 4, please improve the quality of the (a) sub-figure. What are the level of N1-N4? Their curves are not clear.

Reviewer 3 Report

The research developed a methodology to predict yield using optical remote sensing integrating with deep learning technics. A comparison of transfer learning method and data assimilation was tested and the research findings are quite relevant with great applicability.

Additionally  the work is very interesting, current, using modern sensor-systems and state-of-the-art classification algorithms. I consider it an important contribution to the great field of Earth Observation and Agriculture. However, some points could be adjusted.

The introduction is well described and in good context, but more attention to terminology is needed in the area of ​​remote sensing. It wouldn't be too much if there was a little more context and citations.

At the beginning of the methodology, I cannot find the correct citation of the SAAFY model, which should be credited at the first time it appears in the text.

I missed a better conceptual discussion about the PCA (SP-UCI) and the SCE-UA. Who has used it? In what situation? Based on what was it chosen?

 I feel that the reproducibility of the proposed methodology does not seem to be an easy task. It was very economical in the explanation and general discussion of the methods and algorithms. Mainly because the text is not big.

I noticed a certain economy of words and discussions at the end of the conclusion, the text ends very abruptly.

It is necessary to standardize the size and text of the figures.

Except for the small observations, I consider the research to be very relevant with a well-written text.
